# Relationships of Dental Caries and Malocclusion with Oral Health-Related Quality of Life in Lithuanian Adolescents Aged 15 to 18 Years: A Cross-Sectional Study

**DOI:** 10.3390/ijerph17114072

**Published:** 2020-06-08

**Authors:** Aistė Kavaliauskienė, Antanas Šidlauskas, Miglė Žemaitienė, Eglė Slabšinskienė, Apolinaras Zaborskis

**Affiliations:** 1Department of Orthodontics, Faculty of Odontology, Medical Academy, Lithuanian University of Health Sciences, A.Mickevičiaus 9, LT-44307 Kaunas, Lithuania; aiste.kavaliauskiene@lsmuni.lt (A.K.); antanas.sidlauskas@lsmuni.lt (A.Š.); 2Department of Oral Health and Paediatric Dentistry, Medical Academy, Lithuanian University of Health Sciences, A.Mickevičiaus 9, LT-44307 Kaunas, Lithuania; migle.zemaitiene@lsmuni.lt (M.Ž.); egle.slabsinskiene@lsmuni.lt (E.S.); 3Department of Preventive Medicine & Health Research Institute, Faculty of Public Health, Medical Academy, Lithuanian University of Health Sciences, A.Mickevičiaus 9, LT-44307 Kaunas, Lithuania

**Keywords:** oral health-related quality of life, child perceptions questionnaire, dental caries, malocclusion, adolescents, Lithuania

## Abstract

There is a lack of evidence of the moderating effects of caries lesions and malocclusions on oral health-related quality of life (OHRQoL) among older adolescents. This study aimed to evaluate the relationship of dental caries and malocclusion with OHRQoL among Lithuanian adolescents aged 15 to 18 years. A survey in a representative sample of adolescents included a clinical examination to assess dental health status using the DMFT (Decayed, Missing, and Filled Permanent Teeth) index, and malocclusion using the Index of Complexity, Outcome, and Need (ICON). The Child Perceptions Questionnaire (CPQ) was used to evaluate respondents’ OHRQoL. Negative binomial regression was fitted to associate the clinical variables with the CPQ scores. A total of 600 adolescents were examined. The overall mean DMFT score was 2.7. A need for orthodontic treatment was detected among 27.7% of adolescents. Subjects with caries lesions (DMFT > 3) had higher CPQ scores in the domains of functional limitations and social wellbeing (relative risks were 1.35 (95% confidence interval: 1.09–1.67) and 1.30 (1.03–1.64), respectively), while subjects with a need for orthodontic treatment (ICON > 43) had higher CPQ scores in the domains of emotional wellbeing and social wellbeing (relative risks were 1.81 (1.40–2.22), and 1.69 (1.34–2.14), respectively). It was concluded that both dental caries and malocclusion have negative relationships with OHRQoL in adolescents above 15 years, but their effects occur differently in each OHRQoL domain.

## 1. Introduction

Oral-health-related quality of life (OHRQoL) has been the subject of significant research activity and relevant outcome measures for practitioners in the past two decades [1]. OHRQoL is commonly defined as the way patients rate their satisfaction with their current functional and psychosocial state of oral health [2,3]. Generally, it is measured through the use of self-reported instruments. Multiple studies focused on children and adolescents have developed such instruments to assess OHRQoL at specific ages [4,5,6,7]. Among these instruments, the Child Perception Questionnaire (CPQ_11–14_) is the best known and is commonly used in pediatric population studies in many countries [1,8,9]. Ordinarily, the OHRQoL is evaluated by the sum score of the CPQ as a whole and its four domains: OS—oral symptoms, FL—functional limitations, EWB—emotional well-being, and SWB—social wellbeing.

Direct relationships between OHRQoL and oral health conditions have been examined in children and adolescents. Considering the high prevalence of dental caries and malocclusions in children, the impacts of these oral conditions on OHRQoL have received the greatest attention from researchers [1,10]. Other oral conditions, such as gingival and periodontal problems [11,12,13], dental fluorosis [14,15], dental injuries [16,17], and developmental malformations [18], have also been considered. However, these conditions are usually associated with lower levels of prevalence.

Dental caries are one of the most common chronic diseases among children and adolescents and is of increasing concern [19,20]. In a study on the prevalence of dental caries in Lithuania, it was found that the majority of 15 and 18 year old schoolchildren (75%, and 78%, respectively) had experienced dental caries [21]. The effects of dental caries on OHRQoL can involve toothaches and other related problems (poor sleep, inability to concentrate at school, etc.). This hypothesis was tested during the development phase of the CPQ questionnaire. Jokovic et al. [8] found a significant correlation (r = 0.64) between the sum of CPQ scores and the number of caries-damaged teeth among adolescents examined in the clinic. In a population study, Foster Page et al. [22] found that, with an increase in the number of caries-damaged tooth surfaces, the means of the total CPQ questionnaire and OS and SWB scores increased significantly, but there were no significant changes in the FL and EWB domains. The relationship with tooth decay has been tested in almost all CPQ validation studies, but the results have varied. For example, Goursand et al. [23], when testing the validity of the CPQ translation to Portuguese in Brazil, found that the mean scores of all four CPQ domains significantly differed between adolescents with and without dental caries, whereas in analogous studies among Korean teens [24] and Saudi teens [25], a relationship between dental caries intensity and CPQ was only found in the OS domain. In other studies, the association between dental caries intensity and CPQ score was weak or did not reach significance levels [13,26,27,28,29,30].

The number of studies on the impact of malocclusion on OHRQoL in the field of orthodontics is also growing rapidly [31]. There have been many studies demonstrating that orthodontic anomalies can negatively affect the OHRQoL of children and adolescents [31,32,33,34,35]. However, there is a debate in the literature on how these affects depend on the methods of malocclusion assessment (subjective or objective; level of orthodontic treatment need; what index of the orthodontic treatment need was used) [36]. In addition to the above-mentioned obstacles, the heterogeneity of the findings is due to the fact that these studies generally use small samples and they are usually developed in a way that is convenient for researchers [31]. It should also be noted, in evaluating the role of malocclusion, many authors have paid attention to the aesthetic components of teeth and occlusions, suggesting that OHRQoL is more strongly associated with an orthodontic condition observed by children or their parents than objectively diagnosed orthodontic anomalies [36,37,38,39]. Several prior studies have examined the relationship between adolescent malocclusion and OHRQoL, as assessed by the CPQ instrument. Most of these studies concluded that orthodontic anomalies have significant relationships with the EWB and SWB domains of OHRQoL [22,31,32,33,40,41,42]. Only a few studies found an association between malocclusion and the OS and FL domains [10,34]. Sun et al. confirmed this relationship in the case of very severe orthodontic anomalies [10]. Other researchers suggest that such an association can be found among older adolescents, but there is a lack of research to confirm this [31].

While many studies on the influence of different oral conditions on OHRQoL have been conducted, there are few clinical studies with a high level of scientific evidence that evaluated the impacts of both dental caries and malocclusions on OHRQoL [12,13,43,44,45,46]. For most of these studies, the subjects of research were children or adolescents under 14–15 years of age, whereas the association between oral health status and OHRQoL among adolescents above 15 years has rarely been reported in the literature. At the same time, the literature is still inconclusive regarding the comparison of the strength of the examined associations across OHRQoL domains. We hypothesised that both dental caries and malocclusion have negative relationships with OHRQoL, but in different domains. Therefore, the present cross-sectional study aimed to evaluate the relationship impacts of dental caries and malocclusion with oral-health-related quality of life (OHRQoL) among Lithuanian adolescents aged 15 to 18 years.

## 2. Materials and Methods

### 2.1. Ethical Statement

The ethical approval for the examination of schoolchildren was issued by the Kaunas Regional Biomedical Research Ethics Committee on 27 November 2012 (No. BE-2-47). Written informed consent for the children’s examination was obtained from both parents of each child who participated in the study. Permissions were also obtained from the relevant schools and the schoolchildren themselves. The confidentiality and anonymity of the participants was guaranteed.

### 2.2. Sample Size Calculation

The sample size calculation was done using the software G*Power 3.1 (University of Dusseldorf, Dusseldorf, Germany) [47]. This procedure sought to determine the number of subjects that would provide a reliable estimation of the parameters for the association between the CPQ sum score as an outcome variable and oral health variables as determinants. Based on previous studies in adolescent population [22,30], the CPQ sum score was expected to be substantially deviated from the normal distribution. Under these conditions, a Poisson regression method is recommended to assess the relationship between variables [48,49]. Therefore, the calculation procedure assumed the one-sided z tests to be used in a Poisson regression with binomial predictors (π = 0.5), 80% power, a confidence level of 5%, a mean CPQ sum score 10, and a mean sum score ratio (effect size) to be detected of at least 1.1. The minimum number of participants required by these parameters was 260. Since OHRQoL perception is age-dependent [50], two age groups of subjects (15–16 and 17–18 year olds) were selected for the study. Then, the total required sample size was estimated to be *n* = 2 × 260 = 520.

### 2.3. Study Design, Participants, and Data Collection

This observational study had a cross-sectional design and targeted adolescents from 15 to 18 years old, who were divided into two age groups: 15–16 and 17–18.

A random, two-stage sampling design was used to select the initial sample. Twenty public schools were randomly selected from the list of schools in Lithuania, and in each school, classes from Grades 9 to 11 were randomly selected. The initial sample was almost twice as big (*N* = 1018) as the estimated required value, because only half of the schoolchildren were expected to participate in the study.

Data collection was performed in February–May and September–November, 2013. The principals of the selected schools were contacted to introduce the study and discuss the most appropriate circumstances under which to examine the schoolchildren. An informative letter was then sent to the parents and asked them for permission for their child to be examined.

Both a questionnaire survey and a dental examination were conducted to collect data. Under the supervision of a class teacher, the schoolchildren completed anonymous self-administrated questionnaires in the classroom before their dental examinations. In the selected schools, 716 schoolchildren participated in the questionnaire survey, and 649 and 692 of them completed the caries and orthodontic examinations, respectively. A total of 600 subjects, who completed the questionnaire and were examined for both dental caries and malocclusion, were included in the analyses of this study (remaining subjects were absent at school on day of clinical examination or were examined only by one examiner). These subjects represented approximately 60% of the initially selected sample. The main reason (approximately 60%) for non-participation in this study was negative parental consent to participate in the study (Figure 1).

### 2.4. Clinical Examination

Clinical examination of oral health was undertaken by two examiners. Before the fieldwork, one examiner (M.Ž.) was trained in Nyvad’s [51] caries diagnostic criteria, by an expert in this field (E.S.), and was then calibrated (50 subjects were examined in this phase. In regards to different diagnostic criteria, an agreement between examiner and expert (Cohen’s kappa) varied from 0.73 to 0.98). Another examiner (A.K.) was trained and tested for the reliability in accessing orthodontic indices by their author at Cardiff University School of Dentistry (UK, 2012). Two intra-examiner calibration sessions (test-retest of the same 10 subjects in a week period) were performed over the study period. For both examiners, Cohen’s kappa exceeded 0.9 in these sessions. The data obtained during the calibration process were not included in the present statistical analysis.

The schoolchildren were clinically examined in the school’s medical offices, according to the recommendations of the World Health Organization (WHO) for epidemiological surveys [52]. One examiner (M.Ž.) collected data with reference to the dental caries, in order to identify the Decayed, Missing, and Filled Teeth (DMFT) Index for permanent teeth [52]. The dental caries status was numerically expressed by a calculation of the number of decayed, missing, and filled teeth (D +  M  +  F) for each individual. Two groups of adolescents with DMFT scores ≤3 and >3 were formed based on the first global oral health goal announced by WHO for the year 2000 [53].

The other examiner (A.K.) collected data on malocclusion. A quantitative assessment of the severity of the malocclusion was determined using the Index of Complexity, Outcome, and Need (ICON) [54]. This index was based on five occlusal traits, which were weighted and then added together to provide a single ICON score,
ICON = AC × 7+ICON1a × 5 + ICON1b × 5+ICON2 × 5 + ICON3a × 4 + ICON3b × 4 + ICON4 × 3,
where AC was the aesthetic assessment of teeth (scored 1 to 10), ICON1a and ICON1b indicated upper arch crowding or spacing (scored 0 to 5), ICON2 was a cross-bite (0 = no, 1 = yes), ICON3a was an incisor open bite (scored 0 to 4), ICON3b was an incisor overbite (scored 0 to 3), and ICON4 meant a buccal segment with an anteroposterior fit (scores from 0 to 2 were recorded for each of the left and right occlusions and then added). The total score obtained was scaled into two categories for their orthodontic treatment need (score ICON ≤ 43—no orthodontic treatment need and; score ICON > 43—orthodontic treatment need), according to the criteria described by Richmond [54].

### 2.5. OHRQoL Measure

OHRQoL was measured using the Child Perception Questionnaire (CPQ), which was initially developed and validated for children aged 11–14 years in Canada by Jokovic et al. [8]. This instrument has been validated and adopted for Lithuanian adolescents aged 11–14 years [55] and for adolescents up to the age of 18 [56]. The Lithuanian version of this questionnaire displayed good psychometric properties for both age groups of adolescents. The measure within this instrument had a significant association with global life satisfaction [57] and with the need for orthodontic treatment [58].

The CPQ is a 37 item self-reported questionnaire consisting of four health domains: oral symptoms (OS; 6 items), functional limitations (FL; 9 items), emotional wellbeing (EWB; 9 items), and social wellbeing (SWB; 13 items). The items are scored on a 5 point Likert scale ranging from 0 (“never”) to 4 (“every day or almost every day”). For each subject, the sums of the scores were calculated by summing the responses to the items in each domain, and the CPQ sum scores were found by aggregating the scores in all domains. Higher sum scores indicate a worse OHRQoL. In the analyses, the sum scores of the total CPQ and its domains were considered as the outcome (dependent) variables. In the present sample of adolescents aged 15–18, Cronbach’s alpha for the total CPQ was 0.894, indicating good internal consistency reliability.

### 2.6. Other Variables

In the analyses, the data were adjusted for the adolescents’ gender, age group (15–16 and 17–18 years), and family affluence, which seemed to be a significant predictor of OHRQoL in the Lithuanian adolescent population [59]. To evaluate family affluence, the Family Affluence Scale (FAS) was applied [60]. The adolescents were asked how many cars, home computers, children’s bedrooms, and travel experiences/holidays their families had. Based on the respondent’s responses to these four items, a sum score was calculated, and a 3-point ordinal variable FAS was compiled: 1—low affluence (score 0–3); 2—medium affluence (score 4–5); and 3—high affluence (score 6 or higher).

### 2.7. Statistical Analysis

Data were analyzed using SPSS (version 21; IBM SPSS Inc., Chicago, IL, USA, 2012). The normality of the data was tested using the Kolmogorov–Smirnov test. The overall CPQ sum score and domain scores deviated significantly from a normal distribution. Therefore, non-parametric tests were also applied. First, univariate descriptive statistics were used to summarize the subjects’ characteristics. A chi-square test was used for proportional comparisons of the categorical variables among the participants with different demographic characteristics. Second, binary Spearman’s correlations were performed to test the relationships between oral health and OHRQoL measures. Third, the negative binomial regression (NBR) model [48,49] was fitted to identify the effect of oral health conditions on the CPQ sum score. This Poisson regression-based method was chosen for modeling the count data with many zero-valued observations in the CPQ sum score distribution. There is also a presumption that the existing over-dispersion problem can be solved when the negative binomial distribution is used, instead of the more conventional Poisson distribution, and that such an approach helps to better fit the model to the available data [49]. In this model, the sum score of the whole CPQ or its domains was a dependent variable, and the dichotomized DMFT and ICON were independent variables. The data were adjusted for the adolescent age, gender and family affluence. The strength of relationship was evaluated using the ratio of sum score means (RSSM), which indicated how many times the mean value of a dependent variable increased when the value of an independent variable increased by one unit. This measure represented an analogue of the relative risk [49]. The level of statistical significance was set as a *p*-value < 0.05 (two sided).

## 3. Results

A total of 600 adolescents (40.3% boys and 59.7% girls) who participated in the questionnaire survey and all clinical examinations were included in this study. Table 1 shows a summary of the descriptive statistics of these subjects for all the selected variables.

The mean age of the participants was 16.1 ± 1.0 years old; 15–16 year olds represented 60.2% and 17–18 year olds represented 39.8%. Almost half (47.0%) were regarded as living in highly affluent families. The overall mean DMFT score was 2.7 (SD = 2.5). Approximately one-third (31.5%) of the participants had caries experience that exceeded a score of 3 (DMFT > 3). The comparison by gender and age group showed no significant difference in the mean DMFT values. The need for orthodontic treatment was detected in 27.7% of the adolescents (34.7% for boys and 22.9% for girls (*p* = 0.002)).

Table 1 also presents the means and medians for the sum score of the whole CPQ and its domains. The Kolmogorov–Smirnov test and the discrepancy between the mean and median indicated that the distribution of the CPQ sum score deviated substantially from normal and was skewed in the direction of low values. The low values of the median and narrow interquartile ranges showed zero-inflated the CPQ sum score distribution. The median of the whole CPQ sum score did not differ significantly between the age groups. Girls were more concerned about CPQ complaints compared to boys. Therefore, they provided higher CPQ scores (a more detailed description of gender and age influence on the CPQ sum score is presented in the NBR analyses).

The Spearman’s correlation coefficient values shown in Table 2 indicate that the CPQ sum score had a significant positive correlation with the DMTF and ICON sum scores. Increased DMFT and ICON values caused an increased CPQ sum score, resulting in a poorer OHRQoL. Across CPQ domains, DMFT and ICON had the strongest correlations with the EWB domain.

The association between the clinical characteristics of oral health and the CPQ sum score was examined in detail by employing a multivariate NBR analysis (Table 3). In this analysis, the data were adjusted for gender, age, and family affluence. Girls expressed a poorer EWB domain than boys, while age did not significantly influence OHRQoL. Higher family affluence was associated with a lower sum score (RSSM < 1) in all the domains, but a significant association was found with the FL, EWB, and SWB domains, as well as with the CPQ as a whole. Similar to the correlation analysis, the NBR analysis showed that the subjects who needed orthodontic treatment (ICON > 43) had significantly higher CPQ sum scores, resulting in inferior EWB and SWB domains and an inferior overall OHRQoL. Dental caries lesions (DMFT > 3) also had a negative influence on the OHRQoL (RSSM > 1) in all the domains, but this influence was only found to be significant in the FL and EWB domains.

Table 4 presents the results of the analysis of relationships between the clinical characteristics of oral health and the CPQ sum score, which were obtained separately for boys and girls. The comparison between gender groups suggested that these relationships were stronger in girls than boys.

The groups of adolescents (15–16 years and 17–18 years) were compared by age in a similar way (Table 5). This analysis revealed that the negative influence of dental caries lesions on OHRQoL was only significant among adolescents aged 17–18 years in the FL domain. The negative influence of the need for orthodontic treatment on the EWB domain among older adolescents was more noticeable than that among younger adolescents.

## 4. Discussion

The results of the present study demonstrated negative and significant relationship of both the dental caries degree (DMFT index) and malocclusion severity (ICON score) with OHRQoL, and thus, confirmed the main point of our hypothesis. In contrast to many other studies in this field of research [1,9,10,11,31,32,35,61], our study was focused on older adolescents, aged 15 to 18, which is the key novelty of this study.

As an introduction to this study, we have previously performed an analysis in order to evaluate the relationship between orthodontic treatment need and OHRQoL among 11 to 18-year-old adolescents in Lithuania [58]. Three age groups (11–14, 15–16, 17–18) were compared. It was revealed that OHRQoL perception is age-dependent, whereas older adolescents suffered from malocclusions more than their younger counterparts. Therefore, it is important to further understand the relationship between oral/dental health and OHRQoL in older adolescents. However, older adolescence can be considered as a transition from young adolescence (sometimes referred “childhood”) to young adulthood. Adolescents at this age begin to consider their own appearance is important and gain the autonomy to independently request or reject orthodontic treatment [62]. Thus, it is reasonable to assume that persistent, but untreated malocclusions may have psychological and social effects on the OHRQoL of patients of this particular age. However, studies in this field of research often refer to the implications for childhood or adulthood [32,63].

Our previous study [58] also revealed a negative relationship between orthodontic treatment need and OHRQoL predominantly in the domains of emotional and social wellbeing. Additionally, the present study can evaluate the relationship between degree of caries lesions and OHRQoL, as well as comparing it with the relationship between the severity of the malocclusion and OHRQoL. This is the other key novelty of this study.

Epidemiological studies have indicated that tooth decay is an important clinical factor in OHRQoL [8,14,22,23,61,64,65], although some studies have failed to confirm this [13,26,27,28,29,30,31]. In a prospective three-year study conducted on Brazilian adolescents, it was found that tooth decay alone, but not bite problems, had a significant effect on OHRQoL [66]. It is also interesting to note that, in several studies, OHRQoL showed a stronger association not with clinically determined caries intensity, but with subjectively reported caries manifestation [14,67]. Concerning dental caries experience, it is likely that children in the highest disease quartile experience the greatest oral pain and have difficulties in chewing and sleeping; thus, OHRQoL was affected in the OS and FL domains [63,68]. In the present study, however, dental caries status was not associated with the OS domain, which mainly reflects the oral pain aspects of OHRQoL. In agreement with the findings of former studies [63,68], we found a significant negative impact of caries lesions on the FL domain. A more detailed analysis of this effect showed that this impact was common in girls and older adolescents. A social effect related to tooth decay was also detected in the participants in this study. This effect can be explained by the undermining of the adolescents’ social appearance and self-esteem due to altered eating, chewing, or speaking habits [20]. We noticed that this pattern was more prominent in girls.

To date, all the studies have unequivocally confirmed that orthodontic anomalies lead to adverse effects on OHRQoL in children and adolescents. Discussions in the literature are limited to the dimensions of OHRQoL that are most affected and how these effects depend on gender and age [32,36]. Most researchers have found that these effects dominate in the EWB and SWB areas [31,32]. The results of our study were exactly the same. These results can be explained by the fact that social life and emotional feelings are very important for adolescents in shaping their life values [69], and conspicuous orthodontic anomalies often provoke nicknames and bullying [70].

Gender differences were also highlighted in this study. The literature data suggest that women are more likely to seek dental treatment than men [71]. The findings of this study indicated that girls have more difficulties than boys in coping with complaints of tooth decay in the FL and SWB domains, and with complaints of orthodontic anomalies in the EWB and SWB domains. A similar gender difference has been confirmed by other researchers [13,72]. This suggests that girls seek functional, emotional, and social wellbeing more than boys, although they are less likely to be objectively determined to need dental treatment [73]. However, in some studies, these assumptions were not confirmed [74,75]; this phenomenon is thought to be more common in younger children (8–10 years) who have not yet developed gender differences [76].

In general, adolescent age was not found to be a significant factor in this study. However, a comparison between the adolescent groups aged 15–16 years and those aged 17–18 years revealed a strengthened association of OHRQoL with the need for orthodontic treatment in the older age group. In part, this was also observed for the association of the FL domain with dental decay. A slightly different conclusion was reached in a study by Massod et al. [75] that found a weakening of the relationship in older study groups. However, subjects who were older than 18 years were also included in this study, which may have led to their “reconciliation” with orthodontic problems—the longer individuals live with such problems, the less important they become [75].

The application of modern statistical data analysis methods is also one of the advantages of this study. The NBR is the extension of the Poisson regression-based methods, which can be used for modeling the count data that follow various probability distributions other than the normal distribution. It is also useful for modeling the count data with many zero-valued observations in the outcome variable (e.g., CPQ sum score) distribution and to overcome the existing over-dispersion problem [49]. In oral health research, these conditions are quite common and, therefore, the NBR is increasingly chosen to overcome the problems [77,78]. In the current study, NBR was found to fit the data well enough and was apparently better than that of the conventional Poisson regression (data were not shown). However, it is important to compare the alternative models using statistical test and choose the appropriate models accordingly.

While the sample size and methodology of this analysis strengthened its findings, there were several limitations to this study. First, the significant associations found in this study between OHRQoL and the clinical determinants could not be fully explained, considering that the current analysis was based on cross-sectional data, meaning that interpretation of the results could be limited by the difficulty in discriminating between the cause and effect.

Second, a possible weakness of this study may be related to the use of the CPQ, which is a generic measure of OHRQoL. Thus, some of its items could not address aspects specifically related to dental caries or malocclusion [7]. However, the CPQ has been widely used by researchers to evaluate the effects of tooth decay and malocclusion on the OHRQoL of children and adolescents, especially due to the good psychometric properties of the tool [6,7,8,9,22,23,24,25,26,27,28,29,30]. In addition, high reliability of the Lithuanian version of this instrument has been found among adolescents up to the ages of 18 [56], as well as for the present study sample.

Third, an important limitation lies in the measurement of the severity of malocclusion, as only the ICON was chosen for this purpose. To date, there is no evidence-based method for quantification of malocclusion in adolescents, as an assessment tool for all occlusal traits is lacking [79]. Consequently, past studies of the association between OHRQoL and malocclusion in adolescents have been based on different occlusal indices and provided controversial findings; either the strength of the association was weak, or, due to methodological issues, the findings were not conclusive [12].

Finally, a large number of negative parental consents and schoolchildren who were absent on the day of the survey apparently reduced response rate. Therefore, the study may present a selection bias, due to differences between the characteristics of the adolescents participating in the study and those of the non-participants. A low survey response rate can bias results, especially for the univariate distributions of some demographic characteristics (e.g., caries prevalence). However, when examining the relationships between variables in a multivariate analysis, controlling for a variety of background variables, like this study was conducted, there is no evidence of bias from low response rates [80]. It is, nonetheless, true that a sufficiently large number of subjects (600 subjects, 60% of the original number approached) participated in the study. The minimal bias of results was also ensured by other methodological procedures of the study. The external validity was supported by a representative sample of adolescents. The high intra-examiner reliability reinforce the internal validity of the research.

Despite these limitations, we believe that the current findings provide further evidence regarding the impact of dental caries and malocclusion on OHRQoL in older adolescence, which can be considered a transition period from young adolescence to young adulthood. These findings can assist in planning strategies for the improvement of oral health and its related quality of life among adolescents of this particular age.

## 5. Conclusions

This is the first study to measure OHRQoL, and examine its relationship with dental caries and malocclusion, in a representative sample of Lithuanian adolescents aged 15 to 18 years. This study demonstrated a negative relationship between the degree of caries (DMFT index) and OHRQoL, as well as between the severity of the malocclusion (ICON score) and OHRQoL. However, there are indications that these relationships occur differently in each OHRQoL domain. The evaluated relationships were stronger in girls than boys, and they were partially stronger in 17–18 year olds than in 15–16 year olds.

## Figures and Tables

**Figure 1 ijerph-17-04072-f001:**
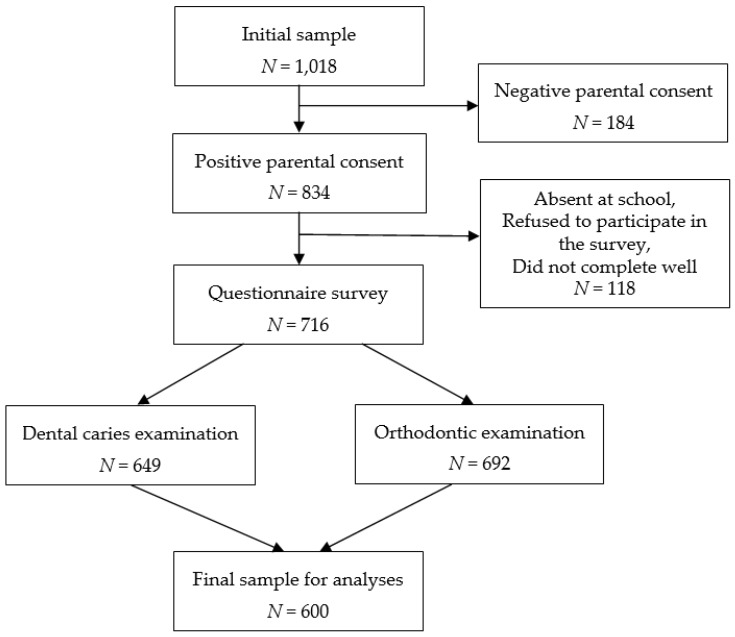
Flow chart of the participant examination process.

**Table 1 ijerph-17-04072-t001:** Descriptive summary statistics of the study subjects (*N* = 600).

Characteristic	*n*	(%)	Mean	(SD)	Median	(IQR)
Gender:						
Boys	242	(40.3)				
Girls	358	(59.7)				
*Age:*			16.1	(1.0)	16	(2)
15–16 years	361	(60.2)				
17–18 years	239	(39.8)				
Family affluence:						
Low	75	(12.5)				
Medium	243	(40.5)				
High	282	(47.0)				
DMFT:			2.7	(2.5)	2	(3)
≤3	411	(68.5)				
>3	189	(31.5)				
ICON:			34.6	(19.7)	29	(26)
≤43	434	(72.3)				
>43	166	(27.7)				
CPQ sum score:						
Whole CPQ			10.7	(10.6)	7	(12)
OS			4.1	(3.0)	3	(4)
FL			2.0	(3.0)	1	(3)
EWB			3.3	(4.9)	2	(4)
SWB			1.3	(3.0))	0	(1)

Note: SD—standard deviation; IQR—interquartile range.

**Table 2 ijerph-17-04072-t002:** Spearman correlation coefficients between the clinical characteristics of oral health and the CPQ sum score (*N* = 600).

Sum Score of Oral Health Characteristics	Sum Score of Whole CPQ and Its Domains				
	Whole CPQ	OS	FL	EWB	SWB
DMFT	**0.16**	0.08	**0.15**	**0.17**	**0.11**
	(*p* < 0.001)	(*p* = 0.051)	(*p* < 0.001)	(*p* < 0.001)	(*p* = 0.005)
ICON	**0.14**	0.05	0.01	**0.20**	**0.13**
	(*p* < 0.001)	(*p* = 0.249)	(*p* = 0.818)	(*p* < 0.001)	(*p* = 0.002)

Note: correlation coefficient values with *p* < 0.05 are in bold.

**Table 3 ijerph-17-04072-t003:** Relationship between the clinical characteristics of oral health and CPQ sum scores: results of the multivariate NBR analysis adjusting data for gender, age, and family affluence (*N* = 600).

Variable	RSSM (95% CI)				
	Whole CPQ	OS	FL	EWB	SWB
*Gender:*					
Boys (ref.)	1.00	1.00	1.00	1.00	1.00
Girls	**1.34**	1.13	**1.23**	**1.92**	1.11
	(*p* = 0.001)	(*p* = 0.216)	(*p* = 0.048)	(*p* < 0.001)	(*p* = 0.365)
	(1.12–1.59)	(0.93–1.36)	(1.00–1.51)	(1.58–2.32)	(0.89–1.40)
*Age:*					
15–16 years (ref.)	1.00	1.00	1.00	1.00	1.00
17–18 years	1.00	1.06	0.85	1.06	0.92
	(*p* = 0.950)	(*p* = 0.569)	(*p* = 0.108)	(*p* = 0.565)	(*p* = 0.466)
	(0.84–1.18)	(0.88–1.27)	(0.69–1.04)	(0.88–1.28)	(0.74–1.15)
*Family affluence:*					
Low (ref.)	1.00	1.00	1.00	1.00	1.00
Medium	0.81	0.96	**0.66**	0.80	**0.66**
	(*p* = 0.122)	(*p* = 0.792)	(*p* = 0.007)	(*p* = 0.127)	(*p* = 0.015)
	(0.62–1.06)	(0.72–1.28)	(0.49–0.89)	(0.59–1.07)	(0.48–0.92)
High	**0.76**	0.92	**0.60**	**0.75**	**0.57**
	(*p* = 0.039)	(*p* = 0.577)	(*p* = 0.001)	(*p* = 0.047)	(*p* = 0.001)
	(0.58–0.99)	(0.69–1.23)	(0.45–0.81)	(0.56–0.99)	(0.41–0.79)
*DMFT:*					
≤3 (ref.)	1.00	1.00	1.00	1.00	1.00
>3	**1.19**	1.14	**1.35**	1.14	**1.30**
	(*p* = 0.049)	(*p* = 0.181)	(*p* = 0.005)	(*p* = 0.212)	(*p* = 0.027)
	(1.00–1.42)	(0.94–1.39)	(1.09–1.67)	(0.93–1.38)	(1.03–1.64)
*ICON:*					
≤43 (ref.)	1.00	1.00	1.00	1.00	1.00
>43	**1.37**	1.11	1.12	**1.81**	**1.69**
	(*p* = 0.001)	(*p* = 0.292)	(*p* = 0.316)	(*p* < 0.001)	(*p* < 0.001)
	(1.14–1.66)	(0.91–1.36)	(0.90–1.40)	(1.48–2.22)	(1.34–2.14)

Note: RSSM—ratio of sum score means; ref.—reference group. Significant values are in bold.

**Table 4 ijerph-17-04072-t004:** Relationship between clinical characteristics of oral health and CPQ sum score by gender: results of the multivariate NBR analysis adjusting data for age and family affluence (*N* = 600).

Variable and Gender Group		RSSM (95% CI)				
		Whole CPQ	OS	FL	EWB	SWB
***DMFT:***						
≤3 (ref.)		1.00	1.00	1.00	1.00	1.00
>3	Boys	1.23	1.19	1.23	1.37	1.24
		(*p* = 0.184)	(*p* = 0.311)	(*p* = 0.272)	(*p* = 0.081)	(*p* = 0.295)
		(0.91–1.68)	(0.85–1.65)	(0.85–1.77)	(0.96–1.94)	(0.83–1.84)
	Girls	1.18	1.12	**1.42**	1.06	**1.38**
		(*p* = 0.157)	(*p* = 0.343)	(*p* = 0.008)	(*p* = 0.616)	(*p* = 0.028)
		(0.94–1.47)	(0.88–1.43)	(1.10–1.85)	(0.84–1.36)	(1.04–1.84)
*ICON:*						
≤43 (ref.)		1.00	1.00	1.00	1.00	1.00
>43	Boys	1.19	1.07	1.11	**1.49**	1.24
		(*p* = 0.233)	(*p* = 0.658)	(*p* = 0.534)	(*p* = 0.015)	(*p* = 0.248)
		(0.90–1.58)	(0.79–1.44)	(0.80–1.56)	(1.08–2.06)	(0.86–1.79)
	Girls	**1.53**	1.15	1.14	**2.02**	**2.22**
		(*p* = 0.001)	(*p* = 0.310)	(*p* = 0.400)	(*p* < 0.001)	(*p* < 0.001)
		(1.19–1.98)	(0.88–1.51)	(0.84–1.53)	(1.54–2.64)	(1.62–3.03)

RSSM—ratio of sum score means; ref.—reference group. Significant values are in bold.

**Table 5 ijerph-17-04072-t005:** Relationship between the clinical characteristics of oral health and the CPQ sum score by age group: results of the multivariate NBR analysis adjusting data for gender and family affluence (*N* = 600).

Variable and Age Group		RSSM (95% CI)				
		Whole CPQ	OS	FL	EWB	SWB
***DMFT:***						
≤3 (ref.)		1.00	1.00	1.00	1.00	1.00
>3	15–16 years	1.20	1.19	1.19	1.19	**1.34**
		(*p* = 0.133)	(*p* = 0.176)	(*p* = 0.207)	(*p* = 0.194)	(*p* = 0.047)
		(0.95–1.51)	(0.93–1.52)	(0.91–1.56)	(0.92–1.53)	(1.01–1.79)
	17–18 years	1.18	1.06	**1.69**	1.08	1.21
		(*p* = 0.267)	(*p* = 0.724)	(*p* = 0.003)	(*p* = 0.623)	(*p* = 0.324)
		(0.88–1.58)	(0.77–1.45)	(1.20–2.39)	(0.79–1.49)	(0.83–1.76)
*ICON:*						
≤43 (ref.)		1.00	1.00	1.00	1.00	1.00
>43	15–16 years	**1.28**	1.08	1.10	**1.59**	**1.71**
		(*p* = 0.042)	(*p* = 0.557)	(*p* = 0.488)	(*p* = 0.001)	(*p* < 0.001)
		(1.01–1.63)	(0.84–1.40)	(0.84–1.46)	(1.22–2.07)	(1.27–2.30)
	17–18 years	**1.52**	1.16	1.18	**2.20**	**1.72**
		(*p* = 0.006)	(*p* = 0.370)	(*p* = 0.366)	(*p* < 0.001)	(*p* = 0.006)
		(1.13–2.06)	(0.84–1.60)	(0.82–1.71)	(1.60–3.04)	(1.17–2.52)

RSSM—ratio of sum score means; ref.—reference group. Significant values are in bold.

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
