# Peer review of "Relationships of Dental Caries and Malocclusion with Oral Health-Related Quality of Life in Lithuanian Adolescents Aged 15 to 18 Years: A Cross-Sectional Study"

_ijerph, 2020, doi:10.3390/ijerph17114072_

Round 1
Reviewer 1 Report
Thank you for the opportunity to review this manuscript, which investigates the association between oral health and oral health related quality of life in older Lithuanian adolescents. The study is a considerable undertaking, involving standardised dental examinations of 600 school children by dental specialists. This represents an impressive body of work and provides valuable insight into the impact of oral disease in adolescents, an age group that is often neglected in medical research broadly. I have a small number of important considerations that the authors should address. Firstly, when conducting such large studies, there is a temptation to conduct multiple analyses and while both malocclusion and dental caries are reasonable exposures to consider, testing the association between oral hygiene and OHRQoL is inappropriate. Poor oral hygiene in itself is not a health condition, and any impact on OHRQoL is likely to be through disease processes like dental caries and gingivitis. As these are investigated in this study already, including oral hygiene detracts from the key message in the article and I recommend removing this part.
Secondly, I note that aspects of this publication have already been published in the same journal previously. It is important to refer to this article at the start of the manuscript and acknowledge the findings so that it is clear how and why this manuscript is different. Where possible, authors should try to avoid breaking apart a single project or research question into multiple papers and so a justification of why this was done in this case is needed.
Finally, the authors highlight the value of this research given the lack of studies of older adolescents. A broader discussion of the importance of this age group/stage of life might be important. For example, studies in childhood often will refer to implications for adulthood. Why is it important to understand OHRQoL in older adolescents?
Some additional minor points are listed below.
Introduction, Page 2, Line 63 - 65
“In a study on the prevalence of dental caries in Lithuania, it was found that the majority of 15 and 18-year-old schoolchildren (75% and 78%, respectively) had dental caries [21].”
Please clarify whether this related to untreated dental carious lesions, or dental caries experienced (including restorations etc).
Introduction Line 65 – 67
“Dental caries complications often lead to tooth decay, which subsequently results in anomalies of the tooth and bite, impaired chewing function, and aesthetic defects [24].
Please review this sentence – the meaning is unclear and it is incorrect.
Introduction Line 108 – 110
“At the same time, the literature is still inconclusive regarding the association between oral health and quality of life among adolescents [49].”
This statement is contradicted by a number of studies including many which are cited earlier in the introduction suggesting that malocclusion can impact on oral health related quality of life.
Methods
Page 4, Line
“These subjects represented approximately 60% of the initial selected sample. The main reason (approximately 60%) for non-participation in this study was negative parental consent to participate in the study.”
Negative parental consent seems to be the most likely reason, what are the other reasons? Absence from school on day of examination? Is there any information available on non-participants to understand the implications of selection bias?
Page 4, Line 151 - 152
“The oral health examination included a calibration process and a clinical stage of data collection, which have both been described previously [21,51].
These references are theses, not peer reviewed journal publications. I suggest instead referring to the publication [ref 61] is more appropriate. I also recommend including this information in the supplementary files.
Author Response
Dear Reviewers,
We thank the reviewers for constructive comments and suggestions, which we have addressed as described below in italic. Changes in the manuscript are with "Track changes" in text (see attached document). Since there were a lot of changes in the paper, we uploaded a new final version of the manuscript, as well as old version corrected using the "Track Changes" function in Microsoft Word, so that changes are easily visible.
To Reviewer #1
Thank you for the opportunity to review this manuscript, which investigates the association between oral health and oral health related quality of life in older Lithuanian adolescents. The study is a considerable undertaking, involving standardised dental examinations of 600 school children by dental specialists. This represents an impressive body of work and provides valuable insight into the impact of oral disease in adolescents, an age group that is often neglected in medical research broadly.
Response: We thank the reviewer for the positive evaluation of our article. We appreciate very much the comments, which are addressed to the first version of the paper.
I have a small number of important considerations that the authors should address. Firstly, when conducting such large studies, there is a temptation to conduct multiple analyses and while both malocclusion and dental caries are reasonable exposures to consider, testing the association between oral hygiene and OHRQoL is inappropriate. Poor oral hygiene in itself is not a health condition, and any impact on OHRQoL is likely to be through disease processes like dental caries and gingivitis. As these are investigated in this study already, including oral hygiene detracts from the key message in the article and I recommend removing this part.
Response: We thank for this important recommendation. All information related to oral hygiene was removed from the text and tables of the paper.
Secondly, I note that aspects of this publication have already been published in the same journal previously. It is important to refer to this article at the start of the manuscript and acknowledge the findings so that it is clear how and why this manuscript is different. Where possible, authors should try to avoid breaking apart a single project or research question into multiple papers and so a justification of why this was done in this case is needed.
Response: We referred to our previous publication [57] in Discussion (lines 313-330) as providing this information in Introduction would have extended this part, which is already too long. This insert describes that the previous study revealed the relevance to study older adolescents, but it was limited because the association between dental caries and OHRQoL has not yet been analyzed. We agree with the recommendation of the second part of this comment but it does not seem to us to be applicable to this article.
Finally, the authors highlight the value of this research given the lack of studies of older adolescents. A broader discussion of the importance of this age group/stage of life might be important. For example, studies in childhood often will refer to implications for adulthood. Why is it important to understand OHRQoL in older adolescents?
Response: We included a broader discussion on this point in lines 315-325 by the way discussing the previous reviewer's comment.
Some additional minor points are listed below.
Introduction, Page 2, Line 63 - 65.
“In a study on the prevalence of dental caries in Lithuania, it was found that the majority of 15 and 18-year-old schoolchildren (75% and 78%, respectively) had dental caries [21].”
Please clarify whether this related to untreated dental carious lesions, or dental caries experienced (including restorations etc).
Response: We clarified this point that studied population "had experienced dental caries" (lines 59-60).
Introduction Line 65 – 67.
“Dental caries complications often lead to tooth decay, which subsequently results in anomalies of the tooth and bite, impaired chewing function, and aesthetic defects [24]."
Please review this sentence – the meaning is unclear and it is incorrect.
Response: This sentence was removed.
Introduction Line 108 – 110. “At the same time, the literature is still inconclusive regarding the association between oral health and quality of life among adolescents [49].”
This statement is contradicted by a number of studies including many which are cited earlier in the introduction suggesting that malocclusion can impact on oral health related quality of life.
Response: This sentence was corrected as follows: "At the same time, the literature is still inconclusive regarding the comparison of strength of the examined associations across OHRQoL domains." (lines 97-98) and, therefore, has link with the idea set out in the paragraph.
Methods
Page 4, Line
“These subjects represented approximately 60% of the initial selected sample. The main reason (approximately 60%) for non-participation in this study was negative parental consent to participate in the study.”
Negative parental consent seems to be the most likely reason, what are the other reasons? Absence from school on day of examination? Is there any information available on non-participants to understand the implications of selection bias?
Response: With regard to this comment, we provided additional information by supplementing the article with a new Figure 1. Flow chart of participant examination process. Unfortunately, detailed information (in rates) on all reasons for non-participation remained unavailable.
Page 4, Line 151 - 152
“The oral health examination included a calibration process and a clinical stage of data collection, which have both been described previously [21,51].
These references are theses, not peer reviewed journal publications. I suggest instead referring to the publication [ref 61] is more appropriate. I also recommend including this information in the supplementary files.
Response: In light of other reviewer comments, we provided more detailed information on the calibration of the examiners (lines 171-179), so we avoided references to other sources about this process in the revised manuscript.

Reviewer 2 Report
The paper shows that ICON score was associated with CPQ score among Lithuanian adolescents aged 15-18 years in a cross-sectional study.
This is an interesting study. However, I would like to make some points regarding the manuscript. The article needs to be revised. Especially, the authors should follow the STROBE guideline and journal’s guideline.
TITLE
1) Please add the study design in the title following the STROBE guideline.
ABSTRACT
1) Please follow the journal’s guideline. The current format is wrong.
INTRODUCTION
1) Please add the hypothesis before the aim.
MATERIALS AND METHODS
1) Please add some parts following the STROBE guideline, such as experimental period, bias, validity, reliability, etc.
2) Why did the authors use Poisson regression in the sample size estimation? It’s not same with the method in this study. Furthermore, there are some unclear parts. Please show the main outcome, appropriate references, the reason why they separate the age group in the sample size estimation.
3) Please add the examiner names and the data of calibration.
4) Why did the authors select the cut-off values; DMFT score = 3, PI = 1 and ICON =43? Please add the details in the text with appropriate references.
5) Did the authors perform the Poisson regression model? How about the results? Are the results similar with those in the NBR? The majority in the CPQ score is zero?! Please show the histogram and the reason why the authors selected NBR, but not other regression models. It is not clear in the current form.
Furthermore, please refer the paper (Minami, M., Lennert-Cody, C. E., Gao, W. and Roman-Verdesoto, M. H. (2007). Modeling shark bycatch: The zero-inflated negative binomial regression model with smoothing, Fisheries Research, 84, 210–221.)
RESULTS
1) Please add the flowchart following the guideline.
2) The authors should add the histograms if appropriate.
3) Please add the data of DMFT, PI and ICON as continuous values in the Table 1.
4) All p values should be shown as the original values, but not P<0.05. Please revise the all tables.
DISCUSSION
1) The discussion part may be changed by the new results.
2) The conclusion is not appropriate. There was only the significant association between total CPQ score and ICON. Please revise the DMFT and PI parts. Furthermore, this is the cross-sectional study. Thus, the authors can’ use “had a negative impact on OHRQOL”.
Author Response
Dear Reviewers,
We thank the reviewers for constructive comments and suggestions, which we have addressed as described below in italic. Changes in the manuscript are with "Track changes" in text (see attached document). Since there were a lot of changes in the paper, we uploaded a new final version of the manuscript, as well as old version corrected using the "Track Changes" function in Microsoft Word, so that changes are easily visible.
To Reviewer #2
The paper shows that ICON score was associated with CPQ score among Lithuanian adolescents aged 15-18 years in a cross-sectional study.
This is an interesting study. However, I would like to make some points regarding the manuscript. The article needs to be revised. Especially, the authors should follow the STROBE guideline and journal’s guideline.
Response: The manuscript has now been reviewed once more and corrected as much as possible in light of the STROBE and journal's guidelines.
TITLE
1) Please add the study design in the title following the STROBE guideline.
Response: Following the change in the paper content and this recommendation, the title was changed to "Relationships of Dental Caries and Malocclusion with Oral Health-Related Quality of Life in Lithuanian Adolescents Aged 15 to 18 Years: A Cross-Sectional Study"
ABSTRACT
1) Please follow the journal’s guideline. The current format is wrong.
Response: The abstract was revised to fulfil STROBE and journal requirements. Examples of abstracts of the papers published in IJERPH were also examined. The abstract was shortened to 228 words.
INTRODUCTION
1) Please add the hypothesis before the aim.
Response: The aim of the study was supplemented with hypothesis "We hypothesised that both dental caries and malocclusion have negative relationship with OHRQoL, but in different domains." (lines 100-102).
MATERIALS AND METHODS
1) Please add some parts following the STROBE guideline, such as experimental period, bias, validity, reliability, etc.
Response: Experimental period (dates of data collection) was indicated (line 129): "Data collection was performed in February–May and September–November, 2013." Other issues have been shortly described in part of study limitations.
2) Why did the authors use Poisson regression in the sample size estimation? It’s not same with the method in this study. Furthermore, there are some unclear parts. Please show the main outcome, appropriate references, the reason why they separate the age group in the sample size estimation.
Response: In statistics, there are many arguments to use a Poison regression in analysis of data that are substantially deviated from the normal distribution. Statistical packages to calculate sample size also include Poisson distribution. We consider this could not to be a limitation, as a Negative binary regression (NBR) that was used in data analysis belongs to the series of methods based on Poisson regression. Appropriate references were also included into revised manuscript, so Sample Size calculation was described as follows (new text is underlined):
"The sample size calculation was done using the software G*Power 3.1 (University of Dusseldorf, Dusseldorf, Germany) [47]. This procedure sought to determine the number of subjects that would provide a reliable estimation of the parameters for the association between the CPQ sum score and oral health variables. Based on previous studies [42,44,48], it was expected that a Poisson regression method should be used to assess the relationship. Therefore, the calculation procedure assumed statistical tests to be used in a Poisson regression with 80% power, a confidence level of 5%, and a mean sum score ratio to be detected of at least 1.1. The minimum number of participants required by these parameters was 260. Since OHRQoL perception is age-dependent [49], two age groups of subjects (15–16 and 17–18 year olds) were selected for the study. Then, the total required sample size was estimated to be n = 2 × 260 = 520."
3) Please add the examiner names and the data of calibration.
Response: The examiner names and the data of calibration were provided in a new paragraph (lines 172-180):
"Clinical examination of oral health was undertaken by two examiners. Before the fieldwork, one examiner (M.Ž.) was trained in Nyvad's [50] caries diagnostic criteria by an expert in this field (E.S.) and was then calibrated (50 subjects were examined in this phase; in regard to different diagnostic criteria, an agreement between examiner and expert (Cohen's kappa) varied from 0.73 to 0.98). Another examiner (A.K.) was trained and tested for the reliability in accessing orthodontic indices by their author at Cardiff University School of Dentistry (UK, 2012). Two intra-examiner calibration sessions (test-retest of the same 10 subjects in a week period) were performed over the study period. For both examiners, Cohen's kappa exceeded 0.9 in these sessions. The data obtained during the calibration process were not included in the present statistical analysis."
4) Why did the authors select the cut-off values; DMFT score = 3, PI = 1 and ICON =43? Please add the details in the text with appropriate references.
Response: These details have been already presented in the previous version of the manuscript (lines: 185-186 and 196-197).
5) Did the authors perform the Poisson regression model? How about the results? Are the results similar with those in the NBR? The majority in the CPQ score is zero?! Please show the histogram and the reason why the authors selected NBR, but not other regression models. It is not clear in the current form.
Furthermore, please refer the paper (Minami, M., Lennert-Cody, C. E., Gao, W. and Roman-Verdesoto, M. H. (2007). Modeling shark bycatch: The zero-inflated negative binomial regression model with smoothing, Fisheries Research, 84, 210–221.)
Response: Before the final decision on which of the statistical methods to use in data analysis of the present study, several methods (linear, logistic, Poisson, NBR and other methods from the Generalized Linear Models) were tested. Although the Poisson regression provided more narrow confidence intervals of parameter estimations, however, goodness of fit of NBR models was apparently better than that of the Poisson regression. For the NBR, the deviation divided by the number of degrees of its freedom was close to 1 (it varied from 0.92 to 1.59). Moreover, the NBR model was helpful to solve the problem of overdisperson that arose due to the highly skewed distribution of the CPQ sum scores. This distribution was zero-inflated and its variance was higher than the mean of the distribution. All these reasons led to the choice of the BNR model. We did not provide all these technical details in the article because we did not want to overload the article with them. We also did not provide a histogram of CPQ sum score because the peculiarities of the CPQ sum score distribution are apparent from the data presented in Table 1. We considered that the statements in the paper recommended are well reflected in the [61] monograph and therefore we did not refer this paper. However, in the light of this comment, we added an appropriate explanation of the NBR option choice:
"This Poisson regression-based method was chosen for modeling the count data with many zero-valued observations in the CPQ sum score distribution. There is also a presumption that the existing overdisperson problem can be solved when the negative binomial distribution is used instead of the more conventional Poisson distribution and that such an approach helps to better fit the model to the available data [61]." (lines 233-237).
RESULTS
1) Please add the flowchart following the guideline.
Response: The flowchart was added in Figure 1.
2) The authors should add the histograms if appropriate.
Response: We considered not to provide a histogram of CPQ sum score because the peculiarities of the CPQ sum score distribution are apparent from the data presented in Table 1
3) Please add the data of DMFT, PI and ICON as continuous values in the Table 1.
Response: We regret that we did not understand this comment. In our opinion, all the main characteristics (mean, standard deviation, median, interquartile range) of the distribution of continuous values were presented in Table 1.
4) All p values should be shown as the original values, but not P<0.05. Please revise the all tables.
Response: Original p values were added in Table 2. There are confidence intervals of the assessments in Tables 3-5, which are more informative characteristics of significance than p values, so we consider there was no need to provide original p values. A such approach is typical in many papers of other authors. Otherwise, showing of original p values will overload these tables.
DISCUSSION
1) The discussion part may be changed by the new results.
Response: We added some new paragraphs and reviewed the remaining text of the discussion in accordance to the new results.
2) The conclusion is not appropriate. There was only the significant association between total CPQ score and ICON. Please revise the DMFT and PI parts. Furthermore, this is the cross-sectional study. Thus, the authors can’ use “had a negative impact on OHRQOL”.
Response: All manuscript was revised in the light of this comment. The main conclusion was re-formulated as follows: "Both dental caries and malocclusion have negative relationships with OHRQoL in adolescents above 15 years, but their effects occur differently in each OHRQoL domain."

Round 2
Reviewer 2 Report
The paper was overall improved. However, there are some issues. The article needs to be revised.
INTRODUCTION
1) It’s ignored; Please add the hypothesis “before the aim.”
MATERIALS AND METHODS
1) Please add some parts following the STROBE guideline, such as experimental period, bias, validity, reliability, etc.
>>> The authors answered “Other issues have been shortly described in part of study limitations.” Where? There are no highlighted areas about bias, validity, reliability, etc.
2) Why did the authors use Poisson regression in the sample size estimation? It’s not same with the method in this study. Furthermore, there are some unclear parts. Please show the main outcome, appropriate references, the reason why they separate the age group in the sample size estimation.
>>> Based on previous studies [42,44,48], what did the authors use as a main outcome to perform sample size estimation? DMFT>3?
RESULTS
1) All p values should be shown as the original values, but not P<0.05. Please revise the all tables.
>>>The authors answered “Response: Original p values were added in Table 2. There are confidence intervals of the assessments in Tables 3-5, which are more informative characteristics of significance than p values, so we consider there was no need to provide original p values. A such approach is typical in many papers of other authors. Otherwise, showing of original p values will overload these tables.” However, when the analysis is different, the meaning of p value is different. Thus, please add the p values.
Author Response
To Reviewer #2
Thank you for re-reviewing our article and appreciating our efforts to improve it. We regret that we were unable to take all of your comments into account at once. We uploaded a previously (in the first round) corrected version of the manuscript using the "Track Changes" function in Microsoft Word to see new corrections in this round. Below, in italic, there are our responses to your comments.
INTRODUCTION
1) It’s ignored; Please add the hypothesis “before the aim.”
Response: The hypothesis was added before the aim. The corrected text can now be read as follows (lines 98–102):
"We hypothesised that both dental caries and malocclusion have negative relationships with OHRQoL, but in different domains. Therefore, the present cross-sectional study aimed to evaluate the relationship impacts of dental caries and malocclusion with oral-health-related quality of life (OHRQoL) among Lithuanian adolescents aged 15 to 18 years."
MATERIALS AND METHODS
1) Please add some parts following the STROBE guideline, such as experimental period, bias, validity, reliability, etc.
>>> The authors answered “Other issues have been shortly described in part of study limitations.” Where? There are no highlighted areas about bias, validity, reliability, etc.
Response: Experimental period was described in line 132: "Data collection was performed in February–May and September–November, 2013."
We wrote down Cronbach's alpha for the total CPQ (lines 218–218): "In the present sample of adolescents aged 15–18, Cronbach's alpha for the total CPQ was 0.894, indicating good internal consistency reliability." The importance of this characteristic is also mentioned in Discussion section (lines 393–394). Other study characteristics (bias, validity, reliability) we described in a new paragraph in Discussion section (lines 402–413):
"Lastly, a large number of negative parental consents and schoolchildren who were absent on the day of the survey apparently reduced response rate. Thus, the study may present a selection bias due to differences between the characteristics of the adolescents participating in the study and those of the non-participants. A low survey response rate can bias results, especially for the univariate distributions of some demographic characteristics (e.g. caries prevalence). But when examining relationships between variables in a multivariate analysis, controlling for a variety of background variables, like this study was conducted, there is no evidence of bias from low response rates [80]. It is, nonetheless, true that a sufficiently large number of subjects (600 subjects, 60% of the original number approached) participated in the study. The minimal bias of results was also ensured by other methodological procedures of the study. The external validity was supported by a representative sample of adolescents. The high intra-examiner reliability reinforce the internal validity of the research."
2) Why did the authors use Poisson regression in the sample size estimation? It’s not same with the method in this study. Furthermore, there are some unclear parts. Please show the main outcome, appropriate references, the reason why they separate the age group in the sample size estimation.
>>> Based on previous studies [42,44,48], what did the authors use as a main outcome to perform sample size estimation? DMFT>3?
Response: We paid particular attention to this comment in the first revision of the manuscript. We hoped that we had substantially amended the text and clarified the ambiguities. In the present revision, we clarified why the Poisson regression was chosen in the sample size estimation. We also indicated the main outcome variable and predictors, as well as provided all information needed for the software G*Power 3.1 to calculate the sample size. Thus, the readers can easily check the accuracy of the calculations using this software.
Really, it's not exactly the same with the method that was used in statistical analysis. Unfortunately, we have to come to terms with this, as there are no more appropriate methods in sample size calculation programs. A Poisson regression was the best one that we could find. On the other hand, we consider this could not to be a serious limitation, as a Negative binary regression (NBR) that was used in data analysis belongs to the series of methods based on Poisson regression. In the discussion part, we have added a separate paragraph on the issue of data analysis (lines 373–382 ).
"The application of modern statistical data analysis methods is also one of the advantages of this study. The NBR is the extension of the Poisson regression-based methods, which can be used for modeling the count data that follow various probability distributions other than the normal distribution. It is also useful for modeling the count data with many zero-valued observations in the outcome variable (e.g. CPQ sum score) distribution and to overcome the existing overdispersion problem [49]. In oral health research, these conditions are quite common and, therefore, the NBR is increasingly chosen to overcome the problems [77,78]. In the current study, NBR was found to fit the data well enough and was apparently better than that of the conventional Poisson regression (data were not shown). However, it is important to compare the alternative models using statistical test and choose the appropriate models accordingly."
In this study we separated two age groups. First, because there is evidence that OHRQoL perception is age-dependent (appropriate reference was included). Second, we were interested whether the strength of the aforementioned relationship depended on the adolescent age. We believe that the choice of the two age groups is properly explained in the text (lines).
"DMFT>3?". Two groups of adolescents with DMFT scores ≤3 and >3 were formed based on the first global oral health goal announced by WHO for the year 2000 (appropriate reference was included). While calculating the sample size, the CPQ predictors were considered to be binomial variables.
RESULTS
1) All p values should be shown as the original values, but not P<0.05. Please revise the all tables.
>>>The authors answered “Response: Original p values were added in Table 2. There are confidence intervals of the assessments in Tables 3-5, which are more informative characteristics of significance than p values, so we consider there was no need to provide original p values. A such approach is typical in many papers of other authors. Otherwise, showing of original p values will overload these tables.” However, when the analysis is different, the meaning of p value is different. Thus, please add the p values.
Response: P-values were also added in Tables 3-5, despite becoming large. We regret that we did not understand the comment ”However, when the analysis is different, the meaning of p value is different". Where is the reviewer seeing a difference in analysis? To our knowledge, p-values are functionally related with the confidence intervals at least in the present analyses.
